# Restoring TRAILR2/DR5-Mediated Activation of Apoptosis upon Endoplasmic Reticulum Stress as a Therapeutic Strategy in Cancer

**DOI:** 10.3390/ijms23168987

**Published:** 2022-08-12

**Authors:** Rocío Mora-Molina, Abelardo López-Rivas

**Affiliations:** Centro Andaluz de Biología Molecular y Medicina Regenerativa-CABIMER, CSIC-Universidad de Sevilla-Universidad Pablo de Olavide, Avda Américo Vespucio 24, 41092 Sevilla, Spain

**Keywords:** apoptosis, extrinsic pathway, TRAILR2/DR5, FLIP, cancer, endoplasmic reticulum stress, unfolded protein response, tumor microenvironment

## Abstract

The uncontrolled proliferation of malignant cells in growing tumors results in the generation of different stressors in the tumor microenvironment, such as nutrient shortage, hypoxia and acidosis, among others, that disrupt endoplasmic reticulum (ER) homeostasis and may lead to ER stress. As a response to ER stress, both normal and tumor cells launch a set of signaling pathways known as the unfolded protein response (UPR) to restore ER proteostasis and maintain cell viability and function. However, under sustained ER stress, an apoptotic cell death process can be induced and this has been the subject of different review articles, although the role of the TRAIL-R2/DR5-activated extrinsic pathway of apoptosis has not yet been thoroughly summarized. In this Review, we provide an updated overview of the molecular mechanisms regulating cell fate decisions in tumor cells undergoing ER stress and discuss the role of the tumor necrosis factor (TNF)-related apoptosis-inducing ligand receptor 2 (TRAIL-R2/DR5) in the final outcome of UPR signaling. Particularly, we focus on the mechanisms controlling cellular FLICE-like inhibitory protein (FLIP) levels in tumor cells undergoing ER stress, which may represent a potential target for therapeutic intervention in cancer.

## 1. Introduction

The endoplasmic reticulum (ER) is a highly dynamic compartment with a wide variety of functions; of these, Ca^2+^ storage and homeostasis or lipid biosynthesis are associated with smooth ER. Nevertheless, a key function of the ER is to control proteostasis, which is mainly linked to rough ER [1]. Indeed, the ER is responsible for at least one-third of all protein synthesis, folding, assembly, trafficking and degradation. Those proteins whose fate is to follow the secretory pathway enter in the ER through the translocon complex. Once in the ER, the nascent protein chains are modified and properly folded by chaperones, peptidylprolyl isomerases, protein disulfide isomerases, oxidoreductases or glycosyltransferases. The oxidizing environment of the ER along with a high concentration of Ca^2+^ specifically promotes the creation of disulfide bonds and chaperone action, respectively. After being correctly folded, proteins are translocated to the Golgi apparatus; then, they are directed to different organelles, the plasma membrane or the extracellular space. Those proteins that are not properly folded or aggregated are sent to the cytosol in order to be ubiquitinated and degraded by the proteasome through a process called ER-associated protein degradation (ERAD) [2]. Briefly, this process begins when ER or cytosolic chaperones recognize unfolded or misfolded motives within a protein, which, subsequently, is retrotranslocated and polyubiquitinated in the cytosol through the action of ubiquitin-activating enzyme (E1), ubiquitin-conjugating enzymes (E2) and ubiquitin ligase (E3) and, finally, degraded by the 26S proteasome. Depending on where the unfolded or misfolded regions are located within the target protein, three different ERAD pathways are involved in the degradation process, ERAD-L, ERAD-M, and ERAD-C, which refer to ERAD substrates with the folding lesion localized in the ER luminal, membrane, or cytosolic-facing domains, respectively [3,4].

Although protein folding and transport are tightly regulated in the ER, there are scenarios, such as a high protein synthesis demand and changes in the Ca^2+^ levels or redox status, that alter ER homeostasis and provoke an excess of misfolded or/and unfolded proteins to appear, a situation known as ER stress. These alterations appear in either physiological (processes accompanied by an increase in protein demand such as proliferation and development of secretory cells, such as plasma B cells or pancreatic β cells) or pathological (hypoxia, inflammation or nutrient deprivation) situations [5,6]. To restore proteostasis upon ER stress, several signaling pathways called unfolded protein response (UPR) become activated. Initially, UPR signaling leads to the inhibition of global protein synthesis and the degradation of unfolded and misfolded proteins. Subsequently, the protein folding capacity of the ER increases by the transcriptional regulation of numerous genes in charge of controlling proteostasis. Some of these genes transcriptionally upregulated through the UPR are *HSPA5*, *CALR*, and *CANX*, which encode the ER-resident chaperones binding-immunoglobulin protein (BiP/GRP78), calnexin and calreticulin, respectively; members of the disulfide isomerase (PDI) family of the ER such as *PDIA3*, *PDIA4*, *PDIA5*, *PDIA6* [7,8,9]; as well as autophagy-related genes (ATG) including *ATG16L1*, *MAP1LC3B*, *ATG12*, *ATG3*, *BCN1*, *ATG7*, *ATG10*, *ATG5* [10]. However, if ER stress is unresolved, the UPR triggers signaling pathways that activate apoptotic cell death through the extrinsic, the intrinsic or both apoptotic pathways. Hence, the UPR determines the cell fate according to the duration and intensity of ER stress [5].

In this review, we summarize what is currently known about signaling pathways induced in cells undergoing ER stress to restore proteostasis and discuss the consequences of sustained ER stress in terms of activation of the apoptotic machinery and the underlying mechanism. We also provide an updated outline of the function of death receptor TRAIL-R2/DR5 in tumor cell fate under ER stress. Finally, as a key player in the control of the extrinsic pathway of apoptosis, we review the role of cellular FLIP proteins in maintaining cell viability under ER stress in cancer cells. This review further elaborates on the deregulation of the mechanisms controlling cFLIPL levels in tumor cells as an essential event in the process leading to apoptosis inhibition under chronic ER stress, which may help us to identify novel therapeutics targets in cancer.

## 2. UPR Signaling Branches

In mammals, the following three main stress sensors are found in the ER membrane: protein kinase RNA (PKR)-like ER kinase (PERK), inositol-requiring protein 1 (IRE1α/β) and activating transcription factor 6 α (ATF6α) (Figure 1). Two models are used to explain how ER stress sensors become activated following ER stress. The first model involves the molecular chaperone BiP/GRP78. Under unstressed situations, these three ER stress sensors are inactive through the binding of BiP/GRP78 to their ER luminal part. When unfolded or misfolded proteins appear, to promote their correct folding, BiP dissociates from ER stress sensors and binds to these proteins with higher affinity. Thus, BiP/GRP78 dissociation from ER stress sensors allows their activation [11]. However, the second mechanism is independent of BiP/GRP78 and implies the direct binding of misfolded proteins to the ER luminal domain of IRE1α and PERK [12,13]. The activation of, at least, IRE1α and PERK after ER stress likely involves both mechanisms [6]. 

### 2.1. IRE1α Pathway

IRE1 is the most conserved arm of the UPR and it was first identified in budding yeast [14]. In mammalian cells, IRE1 is encoded by two genes: *IRE1A* and *IRE1B*, leading to IRE1α and IRE1β protein expression, respectively. While *IRE1A* is constitutively expressed in all cell types, *IRE1B* expression is limited to intestine and lung epithelial cells [15]. Because IRE1α is ubiquitously found, we referred solely to it in this review. IRE1α is a type I transmembrane protein, with cytosolic Ser/Thr kinase and endoribonuclease (RNase) domains. Following BiP dissociation and/or the binding of misfolded proteins at the ER luminal domain [11,12], IRE1α oligomerizes, which allows activation through trans-autophosphorylation. These events create conformational changes that activate the IRE1α endoribonuclease domain, which induces the cleavage of the 26-nucleotide intron from *X-box-binding protein 1 (XBP1)* mRNA. Unspliced XBP1 (XBP1u) protein lacks functional activity and is highly unstable and quickly degraded. In contrast, spliced *XBP1* (*XBP1s*) mRNA codes for a stable transcription factor [16,17]. Once in the nucleus, XBP1s upregulates the expression of genes encoding protein folding and quality control components such as chaperones (calreticulin, calnexin or GRP94) or members of DNAJ/HSP40 family (DNAJA3, DNAJB9 or DNAJC10), which attend HSP70 chaperones stimulating their ATPase activity. Moreover, several ERAD components are also transcriptionally upregulated by XBP1, including ER degradation-enhancing alpha-mannosidase-like protein 1 and 2 (EDEM1 and 2) and some Derlin family members (DERLIN-1 and DERLIN-3) [9]. In addition, XBP1s promotes the transcription of certain genes related to phospholipid synthesis, such as *CHKB* or *GPAT4*, which code for choline kinase and glycerol-3-phosphate acyltransferase 4 enzymes, respectively, in order to expand ER membranes during ER stress [9,18]. These functions of XBP1s are executed in a concerted manner with the ATF6α branch of the UPR [6,15]. IRE1α is also responsible for another process called regulated IRE1-dependent decay (RIDD) in which mRNAs, microRNAs and ribosomal RNAs are degraded through the RNase domain of IRE1α [19,20,21]. As a result of RIDD activation, protein loading in the ER is reduced, helping to restore ER homeostasis. However, excessive RIDD may be harmful and contribute to cell death [2,5]. *XBP1* mRNA splicing or RIDD processes are differentially regulated depending on the oligomerization status of IRE1α [22].

### 2.2. PERK Pathway

Similar to IRE1α, PERK is another type I transmembrane protein with a cytosolic Ser/Thr kinase domain [23]. Once PERK is released from BiP and/or associates with misfolded proteins in its ER luminal domain [11,13], it is activated by oligomerization and trans-autophosphorylation. Then, PERK phosphorylates the alpha subunit of eukaryotic initiator factor 2 (eIF2α) at S51, leading to the inhibition of 5′-cap dependent protein translation [23]. Overall, this mechanism decreases the protein synthesis rate to restore ER homeostasis. However, although general protein translation is prevented, specific mRNAs can still be translated. These mRNAs harbor upstream open reading frames (uORFs), which allow cap-independent translation in stress situations. One of these mRNAs preferentially translated upon ER stress is *activating factor 4 (ATF4)* with two uORFs [24,25]. ATF4 is a basic zipper (bZIP) transcription factor that along with another bZIP dimerization partner controls the transcription of genes related to the antioxidant response, amino acid metabolism or autophagy [10,26]. A key gene regulated by ATF4 is *DDIT3*, which encodes another transcription factor named CCAAT/enhancer-binding protein (C/EBP) homologous protein (CHOP) [27,28]. ATF4 and CHOP can form heterodimers that control the transcription of genes related to protein folding and genes involved in the control of protein synthesis, which may cause proteotoxicity [29]. The genes regulated by the ATF4/CHOP heterodimer include growth arrest and DNA-damage-inducible protein 34 (GADD34)-encoding gene. GADD34 is the regulatory subunit of the protein phosphatase 1 (PP1), which dephosphorylates P-eIF2α restoring protein translation after prolonged ER stress [29,30,31]. 

The responses described thus far are linked to adaptation and cell survival; nevertheless, CHOP also induces the transcription of pro-apoptotic genes, such as *BIM* [32], *TRAILR2/DR5* [33], *PUMA* [34] and *TRB3* [35], and represses the expression of the anti-apoptotic protein BCL-2 during chronic ER stress [36], engaging apoptotic cell death [15]. 

### 2.3. ATF6 Pathway

In contrast to IRE1α and PERK, ATF6α is a type II transmembrane protein harboring a bZIP in its cytosolic domain [37]. Following BiP dissociation (the binding to misfolded proteins has not been attributed to ATF6α), ATF6α moves to the Golgi apparatus where two proteases, serine protease site-1 protease (S1P) and metalloprotease site-2 protease (S2P), cleaves ATF6α, generating the ATF6 p50 N-terminal fragment, which acts as a transcription factor [38,39]. ATF6 p50 promotes the transcription of ERAD-related genes. Furthermore, ATF6 p50 can also function with XBP1s in a heterodimer to mediate the transcription of *XBP1* and genes encoding protein folding enzymes and components of the ERAD pathway [2,5,40].

## 3. UPR Activation: Restore ER Homeostasis or Die in the Attempt

Initially, the activation of UPR signaling pathways aims to restore ER proteostasis to facilitate cell survival. However, unresolved ER stress shifts UPR signaling from adaptation to apoptotic cell death signaling. UPR kinetics can be divided into four phases [41]. First, the immediate response is initiated by decreasing the ER protein load, which occurs by inhibiting protein synthesis and degrading mRNAs through the PERK and IRE1α pathways, respectively. Second, the transcriptional phase allows the upregulation of foldases, chaperones and other proteins related to protein folding in addition to components of ERAD through the PERK, IRE1α and ATF6 pathways. Third, a transitional phase begins in which IRE1α signaling is usually attenuated, while the PERK pathway is maintained, leading to the emergence of pro-apoptotic factors. Finally, sustained ER stress triggers the apoptotic program, and the intrinsic, extrinsic or both apoptotic pathways have been reported to be activated [42,43,44,45]. Therefore, understanding the mechanisms responsible for switching the adaptive response to apoptosis could be a powerful tool for modulating the UPR for clinical application in cancer.

### 3.1. IRE1α Pathway and Apoptosis

In addition to the downstream signaling previously described, upon ER stress, IRE1α serves as a scaffold for the assembly of a platform called the UPRosome at the ER membrane that modulates IRE1α activity and triggers different pathways and responses depending on which proteins are associated (Figure 2) and their downstream signaling [46]. Urano et al. demonstrated that following different ER stress-inducing treatments, IRE1α interacting with TNF-associated factor 2 (TRAF2) was responsible for ER stress-induced Jun amino-terminal kinase (JNK) activation [47]. Consistently, Yang et al. also identified IRE1α as an important factor in JNK activation under ER stress. However, this study described that receptor-interacting protein kinase 1 (RIP1) and tumor necrosis factor receptor 1 (TNFR1), but not TRAF2, are found in the same complex as IRE1α, allowing JNK activation and promoting apoptosis through the intrinsic pathway [48]. In addition to JNK signaling, the UPRosome can lead to nuclear factor-kappa B (NF-κB) activation in tumor cells undergoing ER stress. In this scenario, IRE1α binds TRAF2, which, in turn, recruits the IKK complex. The assembly of this platform allows IKK complex phosphorylation by the IREα kinase domain and the subsequent activation of NF-κB. Finally, NF-κB activation promotes ER stress-induced apoptosis partially due to TNF-α upregulation, which works in an autocrine-manner through TNFR1 [49]. Estornes et al. described the association between IRE1α and RIP1, which indirectly, enhances death receptor-independent caspase-8 activation [50]. The pro-apoptotic BCL-2 family members BAX and BAK, independent of their canonical role at the mitochondria, have also been found to interact with the cytosolic domain of IRE1α during ER stress situations, enhancing IRE1α signaling and helping secretory cells to overcome physiological stresses [51]. Moreover, the BH3-only proteins BIM and PUMA have been identified as regulators of IRE1α endoribonuclease activity, and this regulation occurs through a direct interaction with IRE1α in the UPRosome. While PUMA binding to IRE1α is enhanced during ER stress, the BIM interaction remains unaltered. Both PUMA and BIM are necessary for maintaining or delaying the attenuation of IRE1α RNase activity over the *XBP1* transcript [52]. These are only some examples of IRE1α-binding partners, but many others have been reported [46]. Although IRE1α signaling is usually linked to an adaptive response, excessive or chronic ER stress can lead to prolonged RIDD, which, by degrading chaperone-encoding mRNAs such as *BiP* mRNA, and miRNAs, provokes cell death. Indeed, the degradation of specific miRNAs that prevent *caspase-2* mRNA translation triggers the intrinsic pathway of apoptosis by cleaving BID [21,53]. In addition, miR-17, a miRNA in charge of inhibiting the synthesis of thioredoxin-interacting protein (TXNIP), has been found to be degraded through RIDD. Consequently, TXNIP upregulation causes the assembly of the NLRP3 inflammasome and pyroptotic cell death [54].

### 3.2. PERK Pathway and Apoptosis

PERK-P-eIF2α-ATF4-CHOP is the UPR pathway most linked to persistent ER stress-induced apoptosis (Figure 3). The ATF4/CHOP heterodimer regulates the expression of a battery of target genes with important roles in cell death upon ER stress. One of these genes codes GADD34, the regulatory subunit of protein phosphatase 1 (PP1), which, in a situation of unresolved stress, dephosphorylates P-eIF2α, resulting in the recovery of global protein synthesis, which causes proteotoxicity and ROS production [29]. As previously mentioned, CHOP induces the expression of different pro-apoptotic factors, such as the BH3-only members of the BCL-2 family BIM, PUMA and NOXA, TRAILR2/DR5 and TRB3, and represses the expression of the anti-apoptotic protein BCL-2 [15]. In addition, CHOP has been linked to ROS production in the ER, likely through the transcriptional induction of ER oxidase 1α (ERO1α). This oxidoreductase promotes the formation of disulfide bonds and reactivates protein disulfide isomerases. During these processes, electrons are transferred to O_2_, generating hydrogen peroxide (H_2_O_2_) at the ER lumen. ROS production leads to Ca^2+^ release from the ER via the inositol-1,4,5-triphosphate receptor (IP_3_R). Then, cytosolic Ca^2+^ is taken up by those mitochondria associated with ER membranes, causing ROS production in that organelle, the release of cytochrome *c*, etc. Finally, the cell undergoes apoptotic cell death in a CHOP-dependent manner [2,55].

Recently, the crosstalk between the PERK and IRE1α branches of the UPR has been described [56]. During the adaptive phase of the UPR, both pathways become activated to restore ER proteostasis. The endoribonuclease activity of IRE1α mediates RIDD of *TRAILR2/DR5* mRNA, promoting cell survival. However, after prolonged ER stress, IRE1 signaling is attenuated, and the PERK pathway ultimately leads to apoptosis [57]. During the terminal phase of the UPR, RNA polymerase II-associated protein 2 (RPAP2) phosphatase acts downstream of PERK to attenuate IRE1 signaling, allowing *TRAILR2/DR5* mRNA translation and the execution of apoptosis via the extrinsic pathway [56].

### 3.3. ATF6 Pathway and Apoptosis

The ATF6 branch of the UPR is probably less related to the regulation of apoptosis than the IRE1α and PERK arms. Nevertheless, upon the activation of ATF6 by proteolysis in the Golgi apparatus, ATF6 p50 can bind the *DDIT3* promoter and contribute to regulating the expression of CHOP during ER stress [58], which can provoke the death of the cell by apoptosis. ATF6 has also been shown to induce apoptosis in myoblasts through the downregulation of the anti-apoptotic BCL-2 family member MCL-1. However, a reduction in MCL-1 expression does not occur at the transcriptional level, indicating that ATF6 has an indirect effect on MCL-1 [59].

## 4. Role of the TRAIL-R2-Activated Extrinsic Apoptotic Pathway in the Control of Tumor Progression

Oncogenic transformation switches metabolism to maintain uncontrolled tumor growth. The uncontrolled proliferation of malignant cells and the poor vascularization in growing tumors result in the generation of different stressors in the tumor microenvironment, such as nutrient shortage, hypoxia, acidosis, etc., that disrupt ER homeostasis and cause persistent ER stress in both cancer and stromal cells and the activation of the UPR to fulfil new metabolic requirements [40].

### 4.1. TRAIL-R2 Upregulation in Cells Undergoing ER Stress

In conventional two-dimensional cultures of tumor cells, ER stress-inducing agents have been shown to activate the extrinsic apoptotic pathway through the PERK pathway-mediated induction of the CHOP transcription factor, leading to the upregulation of TRAILR2/DR5 expression [33], which induces the activation of caspase-8 at an intracellular DISC [33,43,57]. Interestingly, DISC assembly under these stressed conditions occurs independently of the TRAIL ligand. Instead, TRAILR2/DR5 clustering is induced by its binding to exposed hydrophobic residues on misfolded proteins at the ER-Golgi intermediate compartment [60]. Collectively, these results demonstrate that the extrinsic pathway of apoptosis plays a relevant role in the terminal UPR. Other stimuli that disrupt protein folding, such as glucose or glutamine deprivation, also cause TRAILR2/DR5 upregulation and caspase-8-dependent cell death [61,62]. As previously mentioned, during the adaptive phase of the UPR, the IRE1α and PERK branches become activated to restore ER homeostasis. Through its RNase activity, IRE1α mediates RIDD of *TRAILR2/DR5* mRNA, which favors cell survival. However, under persistent ER stress, IRE1α signaling is attenuated, and the PERK pathway ultimately leads to apoptotic cell death [57]. Indeed, the attenuation of IRE1α is a consequence of PERK activation. The phosphatase RPAP2 acts downstream of PERK to reduce IRE1α activity, allowing the upregulation of TRAILR2/DR5 and the execution of the apoptotic program [56]. It is likely that IRE1α signaling via RIDD might prevent an early induction of the apoptotic program by cleaving *TRAILR2*/*DR5* mRNA.

### 4.2. Role of Cellular FLICE-like Inhibitory Protein (FLIP) in Apoptosis Regulation upon ER Stress

The FLIP long (FLIP_L_) and FLIP short (FLIP_S_) protein levels play a crucial role in controlling the extrinsic pathway of apoptosis triggered upon TRAILR2/DR5 activation by its ligand [63,64,65,66]. Furthermore, both in vitro and in vivo studies have revealed the survival role of FLIP_L/S_ in the viability of colon cancer cells by inhibiting chemotherapy-induced apoptosis [67]. In addition to the canonical role of FLIP splice isoforms as regulators of DISC-dependent caspase-8 activation at the plasma membrane, it was recently reported that FLIP_L_ localizes to the ER in MEFs, where it was shown to inhibit the caspase-8-mediated cleavage of an ER-localized protein substrate [68]. Interestingly, recent data have revealed that TRAILR2/DR5 upregulation and apoptosis in 2D cultures of colon tumor cells undergoing ER stress are preceded by an early decrease in the protein levels of both FLIP isoforms, which alters the caspase-8/FLIP ratio, facilitating caspase-8 activation at the intracellular DISC and the subsequent induction of apoptosis [69], as has been demonstrated in TRAIL-induced apoptosis [64,65]. Collectively, these results suggest that FLIP proteins play a key role in controlling cell fate decisions upon ER stress in cancer cells (Figure 4).

Multicellular tumor spheroids (MCTSs) closely mimic the properties of solid tumors and represent an intermediate stage between conventional two-dimensional cultures and in vivo models. Growing MCTSs contain a heterogeneous cell population comprising proliferative cells surrounding quiescent cells and a necrotic core [70]. During spheroid growth, the inner layers of cells undergo nutrient and oxygen shortages in addition to the accumulation of cellular waste, causing metabolic changes and leading to a quiescent phenotype in the intermediate layers and cell death in the deepest layers, similar to that observed in solid tumors. Indeed, spheroids beyond a diameter of 500 μm resemble avascular microtumors or micrometastases of cancer patients [71]. MCTSs are markedly more resistant to ER stress than 2D cultures of tumor cells. Interestingly, tumor spheroids maintain the FLIP_L_ levels during persistent ER stress despite activation of the PERK-ATF4-CHOP branch of the UPR and upregulation of the TRAILR2/DR5 protein levels [69]. These data identify the deregulation of the mechanisms controlling the FLIP_L_ levels [72] in spheroid cultures as an essential event in the process leading to apoptosis inhibition under chronic ER stress (Figure 5).

Different studies have indicated that the cell morphology and intracellular signaling pathways are markedly altered in 3D cultures compared to the conventional monolayer cultures of tumor cells [73,74]. It might be speculated that the decline in cell cycle progression resulting from the inhibition of signaling pathways in spheroids [73] may also contribute to maintaining the FLIP levels as previously described in primary T lymphocytes [75]. These cells undergo the downregulation of FLIP protein levels in response to interleukin-2-triggered progression to the S phase of the cell cycle. Hence, by maintaining the FLIP_L_ levels in cell cycle-arrested tumor cells, tumor spheroids might acquire resistance to ER stress-induced caspase-8 activation and apoptosis despite the upregulation of TRAILR2/DR5.

Interestingly, the maintenance of the FLIP_L_ levels in tumor spheroids is associated with increased FLIP_L_ protein stability and resistance to ER stress-induced apoptosis [69]. FLIP proteins are short-lived inhibitory proteins subject to rapid turnover regulated by the ubiquitin–proteasome system [76]. FLIP isoform turnover can be modified by the post-translational modification of FLIP_L/S_ proteins, which alters their stability. In TNF-α-stimulated macrophages, AKT phosphorylates FLIP_L_ at S273, causing a reduction in the FLIP_L_ protein levels [77]. Moreover, in prostate cancer cells, the phosphorylation at T166 in FLIP_L_ is required for its ubiquitination and subsequent proteasomal degradation in response to ROS generation, which enhances the antitumor effect of TRAIL [78]. FLIP_L_ phosphorylation also affects FasL sensitivity in glioma cells, and the differential recruitment of FLIP_L/S_ to the DISC between sensitive and resistant glioma cells has been reported. In the former, FLIP_L/S_ is not detected in FasL-induced DISC, whereas both FLIP isoforms are found in the DISC in resistant tumor cells. Although this study does not address the issue of FLIP_L_ stability, higher expression of FLIP_L/S_, as a consequence of the greater expression and activity of CAMKII, is observed in FasL-resistant glioma cells. Intriguingly, the FLIP_L_ phosphorylated form, generated by CAMKII, is recruited into FasL-induced DISC in resistant glioma cells, generating a phosphorylated p43-FLIPL caspase-8-cleaved fragment. As a result, full caspase-8 processing and activation are impaired, leading to apoptosis inhibition [79]. In addition to phosphorylation, S-nitrosylation may also modify FLIP_L_ proteasomal degradation. In lung epithelial cells, NO donors favor the S-nitrosylation of FLIP_L_ at the caspase-like domain, preventing ubiquitination and further proteasomal degradation, thus conferring resistance to FasL-induced apoptosis [80]. Since FLIP stability can be regulated post-translationally, a proteomic approach comparing 2D and 3D cultures may reveal valuable information regarding the differential post-translational modifications of FLIP_L_, which might be responsible for the higher stability of FLIP_L_ observed in HCT116-derived spheroid cultures undergoing ER stress. Indeed, HCT116-derived spheroids show low AKT activity [73], which might increase FLIP_L_ stability as a result of the decreasing phosphorylation of FLIP_L_ at S273 as previously described in TNF-α-stimulated macrophages [77].

Different ubiquitin E3 ligases have been identified as being responsible for the degradation of FLIP_L/S_ proteins by the proteasome [65,81,82]. Furthermore, the expression of the ubiquitin E3 ligase Itch, which has been reported to target FLIP for degradation [81,83,84,85], is diminished in colorectal carcinoma compared to healthy tissues and adenomas [86]. In gastric cancer, the E3 ubiquitin ligase deltex1 is frequently downregulated. Moreover, in vitro experiments show that deltex1 binds FLIP_L_ and promotes its lysosomal degradation, facilitating TRAIL-induced apoptosis [87]. In addition, the treatment of TRAIL-resistant gastric cancer cells with the antineoplastic agent geldanamycin leads to an increase in the deltex1 levels and a concomitant decrease in the FLIP_L_ levels, rendering these cells sensitive to TRAIL. Thus, deltex1 downregulation in gastric cancer confers TRAIL resistance likely due to the elevated FLIP_L_ levels [87]. Based on these data, Itch, deltex1 or other E3 ubiquitin ligase targeting FLIP proteins might be responsible for the differential regulation of FLIP_L_ between 2D and 3D cellular cultures [82,88,89,90,91].

In several colorectal cancer cell lines, the DNA repair protein Ku70 has been reported to interact with FLIP isoforms mainly in the cytosol, even though Ku70 is mostly found in the nucleus [92]. This interaction does not affect FLIP_L/S_ recruitment to the DISC but increases their stability. Enhancing the acetylation of Ku70 using histone deacetylase inhibitors or mimicking Ku70 acetylation disrupts the interaction between Ku70 and FLIP_L/S_, promoting the polyubiquitination and proteasomal degradation of these latter proteins, which sensitizes cells to TRAILR2/DR5- and caspase-8-dependent apoptosis [92]. Currently, whether the complex between Ku70 and FLIP proteins is increased in tumor spheroids, which would increase FLIP_L/S_ stability by protecting them from proteasomal degradation, is unknown. This situation could lead to the inhibition of the extrinsic apoptotic pathway despite the upregulation of TRAILR2/DR5, which occurs during ER stress in 3D cultures. Importantly, elevated levels of FLIP isoforms have been observed in tumor samples from different cancers, including colorectal tumors, suggesting that this inhibitor plays a protumoral role in the extrinsic apoptotic pathway [93,94,95]. In particular, high FLIP_L_ levels have been found to correlate with a poor prognosis in colorectal cancer patients [96].

## 5. Conclusions

A high growth rate of cancer cells, along with the poor vascularization of tumors, results in stressful conditions in the tumor microenvironment, including low oxygen supply and lack of nutrients, leading to metabolic stress. Metabolic stress can, in turn, adversely affect the environment of the endoplasmic reticulum (ER) and affect the maturation of nascent proteins. The resultant accumulation of the unfolded/misfolded proteins activates the UPR, which serves primarily to protect the cell during stress and helps to restore homeostasis in the ER, facilitating tumor growth. However, if stress is prolonged or there is excessive stimulation of these signalling pathways, TRAILR2/DR5-mediated activation of the extrinsic apoptotic machinery and thereby cell death will occur. In this scenario, recent data suggest that cellular levels of FLIPL may play an important role in tumor cell fate decisions under the stressful conditions of the tumor microenvironment. Thus, in stressful situations, maintaining the levels of this protein that inhibits the extrinsic apoptosis pathway could enable the activation of an adaptive response in tumor cells and other tumor stromal cells, which would promote tumor growth and progression. More importantly, these results also reveal a dependence of tumor cells on maintaining FLIPL levels in the context of the tumor. Therefore, understanding the mechanisms that maintain high levels of FLIP in tumor cells should help in the design of therapeutic strategies that reduce the expression of this protein and, in this way, limit tumor growth. In this sense, it is of vital importance to carry out more studies to define if these mechanisms are specific to tumor cells and may represent a vulnerability of these cells. In the same way, it is necessary to extend these studies to other possible mechanisms of apoptosis signaling through the intrinsic pathway that can be activated in response to stress in the ER, to find out if there are also alterations in tumor cells that can prevent their elimination under the unfavorable conditions of the tumor microenvironment.

## Figures and Tables

**Figure 1 ijms-23-08987-f001:**
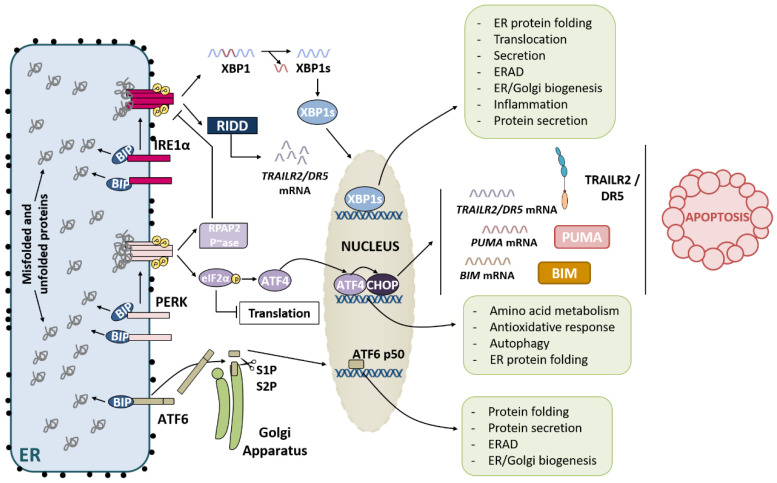
Three ER stress sensors and the induction of the UPR. RIDD: IRE1-α-dependent decay; RPAP2: RNA polymerase II-associated protein 2 phosphatase; S1P: serine protease site-1 protease; S2P: metalloprotease site-2 protease. In mammals, three main stress sensors are found in the ER: Protein kinase RNA (PKR)-like ER kinase (PERK), inositol-requiring protein 1 (IRE1α/β) and activating transcription factor 6 α (ATF6α). Under unstressed situations, all of them are inactive through the binding to BiP in ER lumen. When improperly folded proteins appear, BiP dissociates from ER stress sensors and binds unfolded and misfolded proteins. BiP dissociation from ER stress sensors allows the activation of the unfolded protein response.

**Figure 2 ijms-23-08987-f002:**
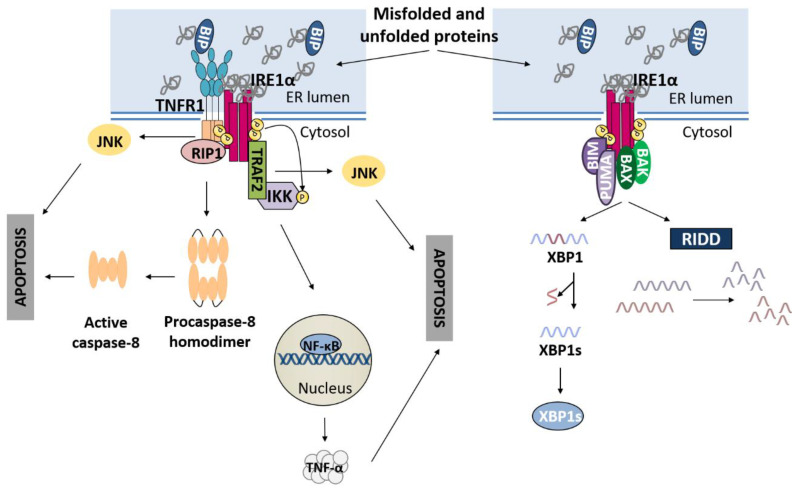
UPRosome platform assembled around IRE1α. Signaling pathways activated following formation of the multi-protein complex named the UPRosome.

**Figure 3 ijms-23-08987-f003:**
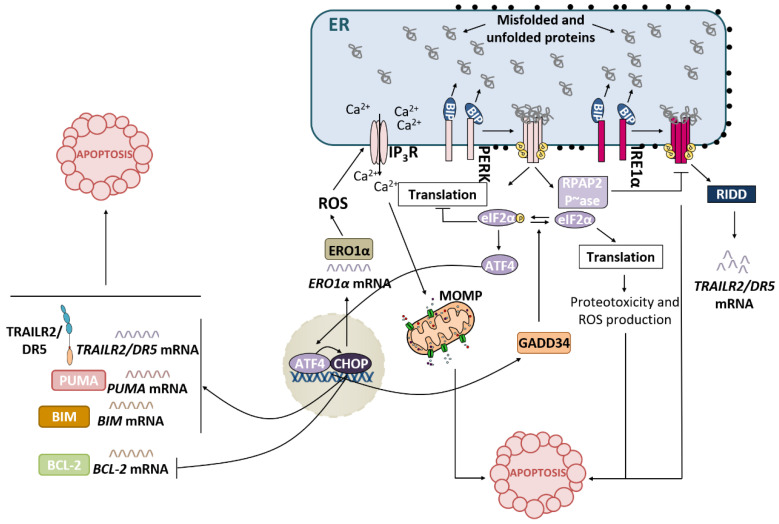
PERK pathway in apoptosis induction. Upon ER stress, PERK activation results in the inhibition of global protein synthesis. At the same time, translation of ATF4 transcription factor will in turn lead to induction of CHOP. ATF4/CHOP heterodimer will be responsible for the upregulation of different genes involved in the control of cell death by apoptosis.

**Figure 4 ijms-23-08987-f004:**
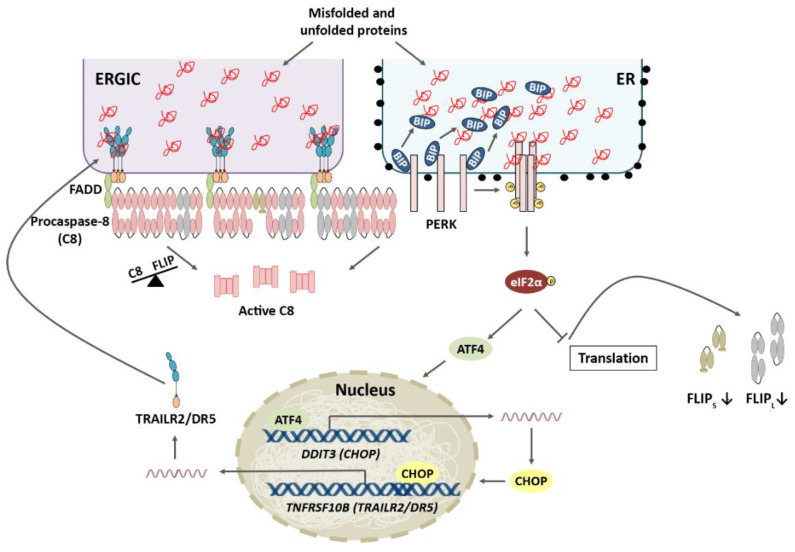
TRAILR2/DR5 upregulation and FLIP downregulation are both required for ER stress-induced apoptosis. ERGIC: ER-Golgi intermediate compartment. CHOP-induced TRAIL-R2/DR5 upregulation and a decrease in the expression levels of cFLIP proteins upon ER stress, which alters the caspase-8/FLIP ratio, facilitate caspase-8 activation and apoptosis.

**Figure 5 ijms-23-08987-f005:**
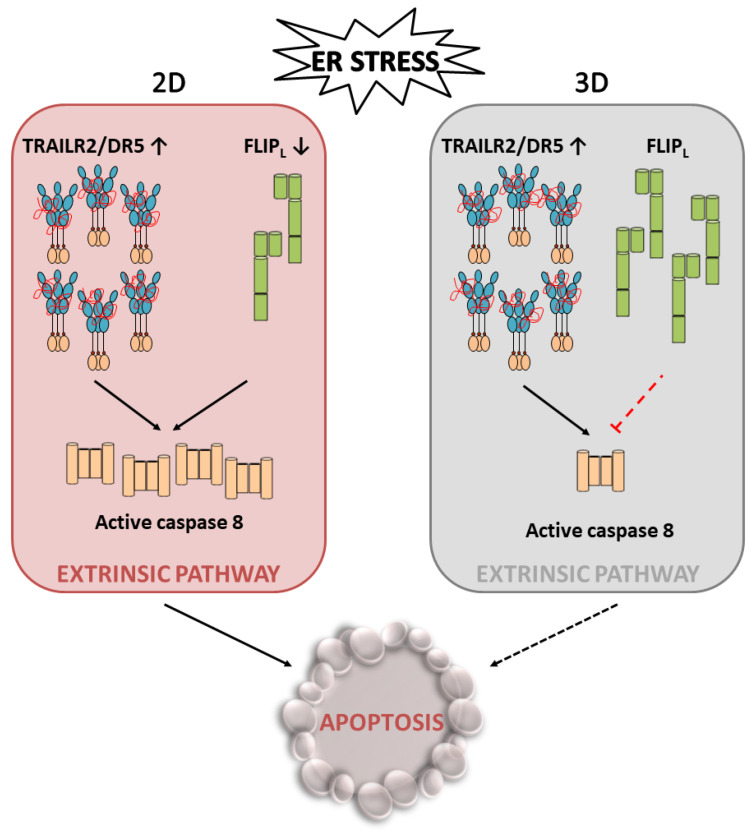
Schematic representation of the differential apoptosis induction between 2D and 3D cell cultures. Despite a similar increase in TRAIL-R2/DR5 levels in 2D and 3D cultures upon ER stress, FLIP levels are maintained in the latter, thus preventing caspase-8 activation of apoptosis.

## Data Availability

Not applicable.

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
