# Peer review of "Restoring TRAILR2/DR5-Mediated Activation of Apoptosis upon Endoplasmic Reticulum Stress as a Therapeutic Strategy in Cancer"

_ijms, 2022, doi:10.3390/ijms23168987_

Round 1
Reviewer 1 Report
This is an interesting and well-written review on TRAILR2/DR5-mediated activation of apoptosis in cancer and the relation to ER stress. It could be improved with the following suggestions:
1) The fonts should be increased in several figures.
2) Brief figure legends, explaining the mechanisms depicted in the figures, should be added, as is customary.
3) Fig. 4 is unclear.
4) P.3 “…protein disulfide isomerases, oxidoreductases or glucosyltransferases.” Should be glycosyltransferases.
5) The English is very good in general. However, there are some typos, a spelling checker should be run. There are also some grammar errors, such as P.4: “…of numerous of genes…”, P.6: “…is prevented, specific mRNA can still be translated.” , P.7: “A key genes regulated by ATF4…”. Grammar should be rechecked and corrected.
Reviewer 2 Report
The literature review is a good introduction to the subject of TRAILR2 / DR5-mediated activation of apoptosis upon endoplasmic reticulum stress as a therapeutic strategy in cancer. Nevertheless, a few corrections need to be made.
1) In the introduction, the authors mention ERAD, please introduce in a few sentences what this process is about.
2) In the introduction, the authors wrote, "Sub Consequently, the protein folding capacity of the ER increases by the transcriptional regulation of numerous of genes in charge of controlling proteostasis. "Can you name examples of these genes?
3) Do figure titles have to be written in bold? All figures contain elements that remain out of focus, in order to improve the reception of the work, please correct these elements.
4) In the subsection IRE1α pathway it says "XBP1s upregulates the expression of genes encoding protein folding, ERAD and protein quality control components”. Can you name these genes?
5) In the subsection IRE1α pathway it says " In addition, XBP1s promotes the transcription of genes related to phospholipid synthesis to expand ER membranes during ER stress.”. Can you name these genes?
Reviewer 3 Report
The effort made by the authors is very valuable, as nowadays it is impossible to follow most of the current literature on a topic of interest. This paper is a comprehensive review focused on the molecular mechanisms controlling cell fate decisions and the role of the TRAIL system in the final outcome of UPR signaling in tumor cells undergoing endoplasmic reticulum stress. The manuscript fits within the scope of the journal. The title is clear and it is adequate to the content of the article. The author’s work on discussing achieved results is appreciated.
I have some recommendations for authors:
The important issue is that the manuscript does not meet the editorial requirements in accordance with the instructions for authors. For this I recommend resubmission.
In the abstract, the purpose of the review does not appear. What did you watch? What important results have you found and what are the directions for future research?
Please include some information about the work method: How do you search literature data? How was your period? Which sources?
What is the novelty and originality of the review, considering that recent articles on this topic are published?
Please include citations for all paragraphs. For example in the introduction chapter, you include only 4 citations. Text citations must be verified throughout the manuscript.
Please improve the quality of the figures!
Complete the conclusions with the personal opinions of the authors regarding the debated topic. What are the limitations observed? What studies need to be continued?
English language and style are fine but minor spell check is required.
Check the bibliography to be written according to the requirements of the journal.
Round 2
Reviewer 3 Report
The authors responded adequately to all my comments with the exception that the manuscript does not meet the editorial requirements in accordance with the instructions for authors. For this I recommend the editor to make the decision.